# Joint Modeling of Content and Discourse Relations in Dialogues

## Abstract

We present a joint modeling approach to identify salient discussion points in spoken meetings as well as to label the discourse relations between speaker turns. A variation of our model is also discussed when discourse relations are treated as latent variables. Experimental results on two popular meeting corpora show that our joint model can outperform SVM-based classifiers for both phrase-based content selection and discourse relation prediction tasks. We also evaluate our model on predicting the consistency among team members' understanding of their group decisions. Classifiers trained with features constructed from our model achieve significant better predictive performance than the state-of-the-art.

## 1 Introduction

Goal-oriented dialogues, such as meetings, negotiations, or customer service transcripts, play an important role in our daily life. Automatically extracting the critical points and important outcomes from dialogues would facilitate generating summaries for complicated conversations, understanding the decision-making process of meetings, or analyzing the effectiveness of collaborations.

We are interested in a specific type of dialogues — spoken meetings, which is a common way for collaboration and idea sharing. Previous work (Kirschner et al., 2012) has shown that discourse structure can be used capture the main discussion points and arguments put forward during problem-solving and decision-making processes in meetings. Indeed, content of different speaker turns do not occur in isolation, and should be interpreted within the context of discourse. Meanwhile, content can also reflect the purpose of speaker turns, thus facilitate with discourse relation understanding. Take the meeting snippet from AMI corpus (Carletta et al., 2006) in Figure 1 as

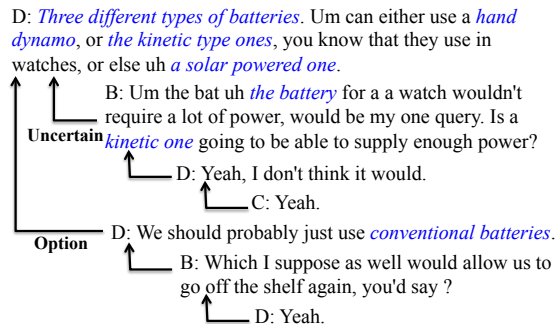

D: *Three different types of batteries*. Um can either use a *hand dynamo*, or *the kinetic type ones*, you know that they use in watches, or else uh *a solar powered one*.

 B: Um the bat uh *the battery* for a a watch wouldn't require a lot of power, would be my one query. Is a
**Uncertain** *kinetic one* going to be able to supply enough power?

 D: Yeah, I don't think it would.

 C: Yeah.

**Option** D: We should probably just use *conventional batteries*.

 B: Which I suppose as well would allow us to go off the shelf again, you'd say ?

 D: Yeah.

Figure 1: A sample clip from AMI meeting corpus. B, C, and D denotes different speakers. Here we highlight salient phrases (in *italics*) that are relevant to the major topic discussed, i.e. "which type of battery to use for the remote control". Arrows indicate discourse structure between speaker turns. We also show some of the discourse relations for illustration.

an example. This discussion is annotated with discourse structure based on the Twente Argumentation Schema (TAS) by Rienks et al. (2005), which focuses on argumentative discourse information. As can be seen, meeting participants evaluate different options by showing doubt (UNCERTAIN), bringing up alternative solution (OPTION), or giving feedback. The discourse information helps with the identification of the key discussion point, i.e., "which type of battery to use", by revealing the discussion flow.

To date, most efforts to leverage discourse information to detect salient content from dialogues have focused on encoding gold-standard discourse relations as features for use in classifier training (Murray et al., 2006; Galley, 2006; McKeown et al., 2007; Bui et al., 2009). However, automatic discourse parsing in dialogues is still a challenging problem (Perret et al., 2016). Moreover, acquiring human annotation on discourse relations is a time-consuming and expensive process, and does not scale for large datasets.

In this paper, we propose *a joint modeling approach to select salient phrases reflecting key dis-*

*cussion points as well as label the discourse relations between speaker turns in spoken meetings.* We hypothesize that leveraging the interaction between content and discourse has the potential to yield better prediction performance on both *phrase-based content selection* and *discourse relation prediction*. Specifically, we utilize argumentative discourse relations as defined in Twente Argument Schema (TAS) (Rienks et al., 2005), where discussions are organized into tree structures with discourse relations labeled between nodes (as shown in Figure 1). Algorithms for joint learning and joint inference are proposed for our model. We also present a variation of our model to treat discourse relations as latent variables when true labels are not available for learning. We envision that the extracted salient phrases by our model can be used as input to abstractive meeting summarization systems (Wang and Cardie, 2013; Mehdad et al., 2014). Combined with the predicted discourse structure, a visualization tool can be exploited to display conversation flow to support intelligent meeting assistant systems.

To the best of our knowledge, our work is the first to jointly model content and discourse relations in meetings. We test our model with two meeting corpora — the AMI corpus (Carletta et al., 2006) and the ICSI corpus (Janin et al., 2003). Experimental results show that our model yields accuracy of 63.2 and 59.2 on phrase selection and discourse prediction on AMI, which is significantly better than comparisons based on Support Vector Machines (SVM) classifiers (accuracy of 57.8 and 51.2). Our model trained with latent discourse also outperforms SVMs on both AMI and ICSI corpora. We further evaluate the usage of selected phrases as extractive meeting summaries. Results evaluated by ROUGE (Lin and Hovy, 2003) demonstrate that our system summaries obtain a ROUGE-SU4 F1 score of 21.3 on AMI corpus, which outperforms two utterance-level extractive summarization baselines that select the longest and the most representative utterance from each discussion.

Moreover, since both content and discourse structure are critical for building shared understanding among participants (Mulder et al., 2002; Mercer, 2004), we further investigate whether our learned model can be utilized to predict the consistency among team members' understanding of their group decisions. This task is first defined as *consistency of understanding* (COU) by Kim and Shah (2016), who have labeled a portion of AMI discussions with consistency or inconsistency labels. We construct features from our model predictions to capture different discourse patterns and word entrainment scores for discussion with different COU level. Results on AMI discussions show that SVM classifiers trained with our features significantly outperform the state-of-the-art results (Kim and Shah, 2016) (F1: 63.1 vs. 50.5) and non-trivial baselines.

## 2 Related Work

Our model is inspired by research work that leverages discourse structure for identifying salient content in conversations, which is still largely reliant on features derived from gold-standard discourse labels (McKeown et al., 2007; Murray et al., 2010; Bokaei et al., 2016). There is much less work that jointly predicts the importance of content along with the discourse structure in dialogus. Oya and Carenini (2014) employs Dynamic Conditional Random Field to recognize sentences in email threads for use in summary as well as their dialogue acts. Only local discourse structures from adjacent utterances are considered. Our model is built on tree structures, which captures more global information.

Our work is also in line with keyphrase identification or phrase-based summarization for conversations. Due to the noisy nature of dialogues, recent work focuses on identifying summary-worthy phrases from meetings (Fernández et al., 2008; Riedhammer et al., 2010) or email threads (Loza et al., 2014). Our work also targets at detecting salient phrases from meetings, but focuses on the joint modeling of critical discussion points and discourse relations held between them.

For the area of discourse analysis in dialogues, a significant amount of work has been done in predicting local discourse structures, such as recognizing dialogue acts or social acts of adjacent utterances from phone conversations (Stolcke et al., 2000; Kalchbrenner and Blunsom, 2013; Ji et al., 2016), spoken meetings (Dielmann and Renals, 2008), or emails (Cohen et al., 2004). Although discourse information from non-adjacent turns has been studied in the context of online discussion forums (Ghosh et al., 2014) and meetings (Hakkani-Tur, 2009), none of them models the effect of discourse structure on content selection, which is a

gap that this work fills in.

## 3 The Joint Model of Content and Discourse Relations

### 3.1 Model Description

Our proposed model learns to jointly perform phrase-based content selection and discourse relation prediction by making use of the interaction between the two sources of information. Assume that a meeting discussion is denoted as $\mathbf{x}$, where $\mathbf{x}$ consists of a sequence of discourse units $\mathbf{x} = \{x_1, x_2, \cdots, x_n\}$. Each discourse unit can be a complete speaker turn or a part of it. As demonstrated in Figure 1, a tree-structured discourse diagram is constructed for each discussion with each discourse unit $x_i$ as a node of the tree. In this work, we consider the argumentative discourse structure by Twente Argument Schema (TAS) (Rienks et al., 2005). For each note $x_i$, it is attached to another node $x_{i'}$ ($i' < i$) in the discussion, and a discourse relation $d_i$ is hold on the link $\langle x_i, x_{i'} \rangle$ ($d_i$ is empty if $x_i$ is the root). Let $\mathbf{t}$ denote the set of links $\langle x_i, x_{i'} \rangle$ in $\mathbf{x}$. Following previous work on discourse analysis in meetings (Rienks et al., 2005; Hakkani-Tur, 2009), we assume that the attachment structure between discourse units are given during both training and testing.

A set of candidate phrases are extracted from each discourse unit $x_i$, from which salient phrases that contain gist information will be identified. We obtain constituent and dependency parses for utterances using Stanford parser (Klein and Manning, 2003). We restrict eligible candidate to be a noun phrase (NP), verb phrase (VP), prepositional phrase (PP), or adjective phrase (ADJP) with at most 5 words, and its head word cannot be a stop word. If a candidate is a parent of another candidate in the constituent parse tree, we will only keep the parent. We further merge a verb and a candidate noun phrase into one candidate if the later is the direct object or subject of the verb. For example, from utterance "let's use a rubber case as well as rubber buttons", we can identify candidates "use a rubber case" and "rubber buttons". For $x_i$, the set of candidate phrases are denoted as $c_i = \{c_{i,1}, c_{i,2}, \cdots, c_{i,m_i}\}$, where $m_i$ is the number of candidates. $c_{i,j}$ takes a value of 1 if the corresponding candidate is selected as salient phrase; otherwise, $c_{i,j}$ is equal to 0. All candidate phrases in discussion $\mathbf{x}$ are represented as $\mathbf{c}$.

We then define a log-linear model with feature parameters $\mathbf{w}$ for the candidate phrases $\mathbf{c}$ and discourse relations $\mathbf{d}$ in $\mathbf{x}$ as:

$$
\begin{aligned}
p(\mathbf{c}, \mathbf{d}|\mathbf{x}, \mathbf{w}) &\propto \exp[\mathbf{w} \cdot \Phi(\mathbf{c}, \mathbf{d}, \mathbf{x})] \\
&\propto \exp[\mathbf{w} \cdot \sum_{i=1, <x_i, x_{i'}>\in \mathbf{t}}^{n} \phi(c_i, d_i, d_{i'}, \mathbf{x})] \\
&\propto \exp[\sum_{i=1, <x_i, x_{i'}>\in \mathbf{t}}^{n} (\mathbf{w_c} \cdot \sum_{j=1}^{m_i} \phi_c(c_{i,j}, \mathbf{x}) \\
&\quad + \mathbf{w_d} \cdot \phi_d(d_i, d_{i'}, \mathbf{x}) + \mathbf{w_{cd}} \cdot \sum_{j=1}^{m_i} \phi_{cd}(c_{i,j}, d_i, \mathbf{x}))]
\end{aligned}
\tag{1}
$$

Here $\Phi(\cdot)$ and $\phi(\cdot)$ denote feature vectors. We utilize three types of feature functions: (1) content-only features $\phi_c(\cdot)$, which capture the importance of phrases, (2) discourse-only features $\phi_d(\cdot)$, which characterize the (potentially higher-order) discourse relations, and (3) joint features of content and discourse $\phi_{cd}(\cdot)$, which model the interaction between the two. $\mathbf{w_c}$, $\mathbf{w_d}$, and $\mathbf{w_c}d$ are corresponding feature parameters. Detailed feature descriptions can be found in Section 3.4.

**Discourse Relations as Latent Variables.** As we mentioned in the introduction, acquiring labeled training data for discourse relations is a time-consuming process since it would require human annotators to inspect the full discussions. Therefore, we further propose a variation of our model where it treats the discourse relations as latent variables, so that $p(\mathbf{c}|\mathbf{x}, \mathbf{w}) = \sum_{\mathbf{d}} p(\mathbf{c}, \mathbf{d}|\mathbf{x}, \mathbf{w})$. Its learning algorithm is slightly different as described in the next section.

### 3.2 Joint Learning for Parameter Estimation

For learning the model parameters $\mathbf{w}$, we employ an algorithm based on SampleRank (Rohanimanesh et al., 2011), which is a stochastic structure learning method. In general, the learning algorithm constructs a sequence of configurations for sample labels as a Markov chain Monte Carlo (MCMC) chain based on a task-specific loss function, where stochastic gradients are distributed across the chain. This is suitable for our learning problem because we aim to optimize the prediction performance for both phrase selection and discourse relations with various types of features.

The full learning procedure is described in Algorithm 1. To start with, the feature weights $\mathbf{w}$ is initialized with each value randomly drawn from $[-1, 1]$. Multiple epochs are run through all samples. For each sample, we randomly initialize the assignment of candidate phrases labels $\mathbf{c}$ and discourse relations $\mathbf{d}$. Then an MCMC chain is con-

structed with a series of configurations $\sigma = (\mathbf{c}, \mathbf{d})$: at each step, it first samples a discourse structure $\mathbf{d}$ based on the proposal distribution $q(\mathbf{d}'|\mathbf{d}, \mathbf{x})$, and then samples phrase labels conditional on the new discourse relations and previous phrase labels based on $q(\mathbf{c}'|\mathbf{c}, \mathbf{d}', \mathbf{x})$. Local search is used for both proposal distributions. The new configuration is accepted if it improves on the score by $\omega(\sigma')$. The parameters $\mathbf{w}$ are updated accordingly.

For the scorer $\omega$, we use a weighted combination of F1 scores of phrase selection ($F1_c$) and discourse relation prediction ($F1_d$): $\omega(\sigma) = \alpha \cdot F1_c + (1 - \alpha) \cdot F1_d$. We fix $\alpha$ to 0.1.

When discourse relations are treated as latent, we initialize discourse relations for each sample with a label in $\{1, 2, \ldots, K\}$ if there are $K$ relations indicated, and we only use $F1_c$ as the scorer.

---

**Input** : $\mathbf{X} = \{\mathbf{x}\}$: discussions in the training set,
$\eta$: learning rate, $\epsilon$: number of epochs,
$\delta$: number of sampling rounds,
$\omega(\cdot)$: scoring function, $\Phi(\cdot)$: feature functions
**Output**: feature weights $\frac{1}{|\mathcal{W}|} \sum_{\mathbf{w} \in \mathcal{W}} \mathbf{w}$

Initialize $\mathbf{w}$;
$\mathcal{W} \leftarrow \{\mathbf{w}\}$;
**for** $e = 1$ *to* $\epsilon$ **do**
 **for** $\mathbf{x}$ *in* $\mathbf{X}$ **do**
 // Initialize configuration for $\mathbf{x}$
 Initialize $\mathbf{c}$ and $\mathbf{d}$;
 $\sigma = (\mathbf{c}, \mathbf{d})$;
 **for** $s = 1$ *to* $\delta$ **do**
 // New configuration via local search
 $\mathbf{d}' \sim q_d(\cdot|\mathbf{x}, \mathbf{d})$;
 $\mathbf{c}' \sim q_d(\cdot|\mathbf{x}, \mathbf{c}, \mathbf{d}')$;
 $\sigma' = (\mathbf{c}', \mathbf{d}')$;
 $\sigma^+ = \arg\max_{\tilde{\sigma} \in \{\sigma, \sigma'\}} \omega(\tilde{\sigma})$;
 $\sigma^- = \arg\min_{\tilde{\sigma} \in \{\sigma, \sigma'\}} \omega(\tilde{\sigma})$;
 $\hat{\nabla} = \Phi(\sigma^+) - \Phi(\sigma^-)$;
 $\Delta\omega = \omega(\sigma^+) - \omega(\sigma^-)$;
 // Update parameters
 **if** $\mathbf{w} \cdot \hat{\nabla} < \Delta\omega$ & $\Delta\omega \neq 0$ **then**
 $\mathbf{w}' = \mathbf{w} + \eta \cdot \hat{\nabla}$;
 Add $\mathbf{w}'$ in $\mathcal{W}$;
 **end**
 // Accept or reject new configuration
 **if** $\sigma^+ == \sigma'$ **then**
 $\sigma = \sigma'$
 **end**
 **end**
 **end**
**end**

**Algorithm 1:** SampleRank-based joint learning.

---

### 3.3 Joint Inference for Prediction

Given a new sample $\mathbf{x}$ and learned parameters $\mathbf{w}$, we predict phrase labels and discourse relations as $\arg\max_{\mathbf{c}, \mathbf{d}} p(\mathbf{c}, \mathbf{d}|\mathbf{x}, \mathbf{w})$.

Dynamic programming can be employed to carry out joint inference, however, it would be time-consuming since our objective function has a large search space for both content and discourse labels. Hence we propose an alternating optimizing algorithm to search for $\mathbf{c}$ and $\mathbf{d}$ iteratively. Concretely, for each iteration, we first optimize on $\mathbf{d}$ by maximizing $\sum_{i=1, <x_i, x_{i'}> \in \mathbf{t}} (\mathbf{w_d} \cdot \phi_d(d_i, d_{i'}, \mathbf{x}) + \mathbf{w_{cd}} \cdot \sum_{j=1}^{m_i} \phi_{cd}(c_{i,j}, d_i, \mathbf{x}))$. Message-passing (Smith and Eisner, 2008) is used to find the best $\mathbf{d}$.

In the second step, we search for $\mathbf{c}$ that maximizes $\sum_{i=1, <x_i, x_i'> \in \mathbf{t}} (\mathbf{w_c} \cdot \sum_{j=1}^{m_i} \phi_c(c_{i,j}, \mathbf{x}) + \mathbf{w_{cd}} \cdot \sum_{j=1}^{m_i} \phi_{cd}(c_{i,j}, d_i, \mathbf{x}))$. We believe that candidate phrases based on the same concepts should have the same predicted label. Therefore, candidates of the same phrase type and sharing the same head word are grouped into one cluster. We then cast our task as an integer linear programming problem.[1] We optimize our objective function under constraints: (1) $c_{i,j} = c_{i',j'}$ if $c_{i,j}$ and $c_{i',j'}$ are in the same cluster, and (2) $c_{i,j} \in 0, 1, \forall i, j$.

The inference process is the same for models trained with latent discourse relations.

### 3.4 Features

We use features that characterize content, discourse relations, and the combination of both.

**Content Features.** For modeling the salience of content, we calculate the minimum, maximum, and average of `TF-IDF` scores of words and `number of content words` in each phrase based on the intuition that important phrases tend to have more content words with high TF-IDF scores (Fernández et al., 2008). We also consider whether the head word of the phrase has been `mentioned in preceding turn`, which implies the focus of a discussion. The `size of the cluster` this phrase belongs to is also included. `Number of POS tags` and `phrase types` are counted to characterize the syntactic structure. Previous work (Wang and Cardie, 2012) has found that a discussion usually ends with decision-relevant information. We thus identify the `absolute and relative positions` of the turn containing the candidate phrase in the discussion. Finally, we record whether the candidate phrase is `uttered by the main speaker`, who speakers the most words in the discussion.

**Discourse Features.** For each discourse unit, we

---

[1] We use lpsolve: http://lpsolve.sourceforge.net/5.5/.

collect the `dialogue act types` of the current unit and its parent node in discourse tree, whether there is any `adjacency pair` held between the two nodes (Hakkani-Tur, 2009), and the `Jaccard similarity` between them. We record whether two turns are `uttered by the same speaker`, for example, ELABORATION is commonly observed between the turns from the same participant. We also calculate the `number of candidate phrases` based on the observation that OPTION and SPECIALIZATION tend to contain more informative words than POSITIVE feedback. Length of the discourse unit is also relevant. Therefore, we compute the `time span` and `number of words`. To incorporate global structure features, we encode the `depth of the node` in the discourse tree and `the number of its siblings`. Finally, we include an `order-2 discourse relation` feature that encodes the relation between current discourse unit and its parent, and the relation between the parent and its grandparent if it exists.

**Joint Features.** For modeling the interaction between content and discourse, the discourse relation is added to each content feature to compose a joint feature. For example, if candidate $c$ in discussion $x$ has a content feature $\phi_{[avg-TFIDF]}(c, \mathbf{x})$ with a value of 0.5, and its discourse relation $d$ is POSITIVE, then the joint feature takes the form of $\phi_{[avg-TFIDF, Positive]}(c, d, \mathbf{x}) = 0.5$.

## 4 Datasets and Experimental Setup

**Meeting Corpora.** We evaluate our joint model on two meeting corpora with rich annotations: the AMI meeting corpus (Carletta et al., 2006) and the ICSI meeting corpus (Janin et al., 2003). AMI corpus consists of 139 scenario-driven meetings, and ICSI corpus contains 75 naturally occurring meetings. Both of the corpora are annotated with dialogue acts, adjacency pairs, and topic segmentation. We treat each topic segment as one discussion, and remove discussions with less than 10 turns or labeled as "opening". 694 discussions from AMI and 1139 discussions from ICSI are extracted, and these two datasets are henceforth referred as AMI-FULL and ICSI-FULL.

**Acquiring Gold-Standard Labels.** Both corpora contain human constructed abstractive summaries and extractive summaries on meeting level. Short abstracts, usually in one sentence, are constructed by meeting participants — *participant summaries*, and external annotators — *abstractive summaries*. Dialogue acts that contribute to important output of the meeting, e.g. decisions, are identified and used as extractive summaries, and some of them are also linked to the corresponding abstracts.

Since the corpora do not contain phrase-level importance annotation, we induce gold-standard labels for candidate phrases based on the following rule. A candidate phrase is considered as a positive sample if its head word is contained in any abstractive summary or participant summary.

Furthermore, a subset of discussions in AMI-FULL are annotated with discourse structure and relations based on Twente Argumentation Schema (TAS) by Rienks et al. (2005)[2]. A tree-structured argument diagram (as shown in Figure 1) is created for each discussion or a part of the discussion. The nodes of the tree contain partial or complete speaker turns, and discourse relation types are labeled on the links between the nodes. In total, we have 129 discussions annotated with discourse labels. This dataset is called AMI-SUB hereafter.

**Experimental Setup.** 5-fold cross validation is used for all experiments. All real-valued features are uniformly normalized to [0,1]. For the joint learning algorithm, we use 10 epochs and carry out 50 sampling for MCMC for each training sample. The learning rate is set to 0.01. We run the learning algorithm for 20 times, and use the average of the learned weights as the final parameter values. For models trained with latent discourse relations, we fix the number of relations to 9.

**Baselines and Comparisons.** For both phrase-based content selection and discourse relation prediction tasks, we consider (1) a baseline that always predicts the majority label (Majority), and (2) a random baseline. Previous work has shown that Support Vector Machines (SVMs)-based classifiers achieve state-of-the-art performance for keyphrase selection in meetings (Fernández et al., 2008; Wang and Cardie, 2013) and discourse parsing for formal text (Hernault et al., 2010). Therefore, we compare with linear SVM-based classifiers, trained with the same feature set of content features or discourse features. We fix the trade-off parameter to 1.0 for all SVM-based experiments. For discourse relation prediction, we use one-vs-rest strategy to build multiple binary classifiers.

---

[2]There are 9 types of relations in TAS: POSITIVE, NEGATIVE, UNCERTAIN, REQUEST, SPECIALIZATION, ELABORATION, OPTION, OPTION EXCLUSION, and SUBJECT-TO.

| | Acc | F1 |
|---|---|---|
| **Comparisons** | | |
| Baseline (Majority) | 60.1 | 37.5 |
| Baseline (Random) | 50.0 | 33.1 |
| SVM (w content features in § 3.4) | 57.8 | 54.6 |
| **Our Models** | | |
| Joint-Learn + Joint-Inference | **63.2**∗ | **62.6**∗ |
| Joint-Learn + Separate-Inference | 57.9 | 57.8 |
| Separate-Learn | 53.4 | 52.6 |
| **Our Models (Latent Discourse)** | | |
| *w/ True Attachment Structure* | | |
| Joint-Learn + Joint-Inference | 60.3∗ | 60.3∗ |
| Joint-Learn + Separate-Inference | 56.4 | 56.2 |
| *w/o True Attachment Structure* | | |
| Joint-Learn + Joint-Inference | 56.4 | 56.4 |
| Joint-Learn + Separate-Inference | 52.7 | 52.3 |

Table 1: Phrase-based content selection performance on AMI-SUB with accuracy (acc) and F1. We display results of our models trained with gold-standard discourse relation labels and with latent discourse relations. For the later, we also show results based on *True Attachment Structure*, where the gold-standard attachments are known, and without the *True Attachment Structure*. Our models that significantly outperform SVM-based model are highlighted with ∗ ($p < 0.05$, paired $t$-test). Best result for each column is in **bold**.

| | Acc | F1 |
|---|---|---|
| **Comparisons** | | |
| Baseline (Majority) | 51.2 | 7.5 |
| Baseline (Random) | 11.1 | 1.9 |
| SVM (w discourse features in § 3.4) | 51.2 | 22.8 |
| **Our Models** | | |
| Joint-Learn + Joint-Inference | 58.0∗ | 21.7 |
| Joint-Learn + Separate-Inference | **59.2**∗ | 23.4 |
| Separate-Learn | 58.2∗ | **25.1** |

Table 2: Discourse relation prediction performance on AMI-SUB. Our models that significantly outperform SVM-based model are highlighted with ∗ ($p < 0.05$, paired $t$-test).

## 5 Experimental Results

### 5.1 Phrase Selection and Discourse Labeling

Here we present the experimental results on phrase-based content selection and discourse relation prediction. We experiment with two variations of our joint model: one is trained on gold-standard discourse relations, the other is trained by treating discourse relations as latent models as described in Section 3.1. Remember that we have gold-standard argument diagrams on the AMI-SUB dataset, we can thus conduct experiments by assuming the *True Attachment Structure* is given for latent versions. When argument diagrams are not available, we build a tree among the turns in each discussion as follows. Two turns are attached if there is any adjacency pair between them. If one turn is attached to more than one previous turns, the closest one is considered. For the rest of the turns, they are attached to the preceding turn.

We also investigate whether joint learning and joint inference can produce better prediction performance. We consider joint learning with separate inference, where only content features or discourse features are used for prediction. We further study learning separate classifiers for content selection and discourse relations without joint features (Separate-Learn).

We first show the phrase selection and discourse relation prediction results on AMI-SUB in Tables 1 and 2. As shown in Table 1, our models, trained with gold-standard discourse relations or latent ones with true attachment structure, yield significant better accuracy and F1 scores than SVM-based classifiers trained with the same feature sets for phrase selection (paired $t$-test, $p < 0.05$). Moreover, both Tables 1 and 2 demonstrate that joint learning usually produces superior performance for both tasks than separate learning. Combined with joint inference, our model obtains the best accuracy and F1 on phrase selection. This indicates that leveraging the interplay between content and discourse boost the prediction performance. Similar results are achieved on AMI-FULL and ICSI-FULL in Table 3, where latent discourse relations without true attachment structure are employed for training.

| | AMI-FULL | | ICSI-FULL | |
|---|---|---|---|---|
| | Acc | F1 | Acc | F1 |
| **Comparisons** | | | | |
| Baseline (Majority) | 61.8 | 38.2 | **75.3** | 43.0 |
| Baseline (Random) | 50.0 | 32.9 | 50.0 | 31.4 |
| SVM (with content features in § 3.4) | 58.6 | 56.7 | 66.2 | 53.1 |
| **Our Models (Latent Discourse)** | | | | |
| Joint-Learn + Joint-Inference | **63.4**∗ | **63.0**∗ | 73.5∗ | 61.4∗ |
| Joint-Learn + Separate-Inference | 57.7 | 57.5 | 70.0∗ | **62.7**∗ |

Table 3: Phrase-based content selection performance on AMI-FULL and ICSI-FULL. We display results of our models trained with latent discourse relations. Results that are significantly better than SVM-based model are highlighted with ∗ ($p < 0.05$, paired $t$-test).

### 5.2 Phrase-Based Extractive Summarization

We further evaluate whether the prediction of the content selection component can be used for summarizing the key points on discussion level. For each discussion, salient phrases identified by our model are concatenated in sequence for use as the summary. We consider two types of gold-standard summaries. One is utterance-level extractive summary, which consists of human labeled summary-worthy utterances. The other is abstractive summary, where we collect human abstract with at least one link from summary-worthy utterances.

| *Extractive Summaries as Gold-Standard* | | | | | | |
|---|---|---|---|---|---|---|
| | ROUGE-1 | | | ROUGE-SU4 | | |
| | Len | Prec | Rec | F1 | Prec | Rec | F1 |
| Longest DA | 30.9 | 64.4 | 15.0 | 23.1 | 58.6 | 9.3 | 15.3 |
| Centroid DA | 17.5 | **73.9** | 13.4 | 20.8 | **62.5** | 6.9 | 11.3 |
| SVM | 49.8 | 47.1 | 24.1 | 27.5 | 22.7 | 10.7 | 11.8 |
| Our Model | 66.6 | 45.4 | 44.7 | 41.1* | 24.1* | 23.4* | 20.9* |
| Our Model-latent | 85.9 | 42.9 | **49.3** | **42.4**∗ | 21.6 | **25.7**∗ | **21.3**∗ |
| *Abstractive Summaries as Gold-Standard* | | | | | | |
| | ROUGE1 | | | ROUGE-SU4 | | |
| | Len | Prec | Rec | F1 | Prec | Rec | F1 |
| Longest DA | 30.9 | 14.8 | 5.5 | 7.4 | 4.8 | 1.4 | 1.9 |
| Centroid DA | 17.5 | **24.9** | 5.6 | 8.5 | **11.6** | 1.4 | 2.2 |
| SVM | 49.8 | 13.3 | 9.7 | 9.5 | 4.4 | 2.4 | 2.4 |
| Our Model | 66.6 | 12.6 | 18.9 | **13.1**∗ | 3.8 | 5.5* | **3.7**∗ |
| Our Model-latent | 85.9 | 11.4 | **20.0** | 12.4* | 3.3 | **6.1**∗ | 3.5* |

Table 4: ROUGE scores for phrase-based extractive summarization evaluated against human-constructed utterance-level extractive summaries and abstractive summaries. Our models that statistically significantly outperform all comparisons are highlighted with ∗ ($p < 0.05$, paired $t$-test).

We calculate scores based on ROUGE (Lin and Hovy, 2003), which is a popular tool for evaluating text summarization (Gillick et al., 2009; Liu and Liu, 2010). ROUGE-1 (unigrams) and ROUGE-SU4 (skip-bigrams with at most 4 words in between) are used. Following previous work on meeting summarization (Riedhammer et al., 2010; Wang and Cardie, 2013), we consider two dialogue act-level summarization baselines: (1) LONGEST DA in each discussion is selected as the summary, and (2) CENTROID DA, the one with the highest TF-IDF similarity with all DAs in the discussion. We also compare with summaries consisting of salient phrases predicted by an SVM classifier trained with on our content features.

From the results in Table 4, we can see that phrase-based extractive summarization methods by our system and SVM-based classifiers can yield better ROUGE scores for recall and F1 than baselines that extract the whole sentences. Meanwhile, our system significantly outperforms the SVM-based classifiers when evaluated on ROUGE recall and F1, while achieving comparable precision.

Sample summaries by our model along with two baselines are displayed in Figure 2. Utterance-level extract-based baselines unavoidably contain disfluency and unnecessary details. Our phrase-based extractive summary is able to capture the key points from both the argumentation process and important outcomes of the conversation.

## 5.3 Further Analysis and Discussions

**Features Analysis.** We first discuss salient features with top weights learned by our joint model.

| **Meeting Clip**: |
|---|
| D: can we uh power a light in this? can we get a strong enough battery to power a light? |
| A: um i think we could because the lcd panel requires power, and the lcd is a form of a light so that. . . |
| D: . . . it's gonna have to have something high-tech about it and that's gonna take battery power. . . |
| D: well m i'm thinking along the lines of you're you're in the dark watching a dvd and you um you find the thing in the dark and you go like this . . . oh where's the volume button in the dark, and uh y you just touch it . . . and it lights up or something. |

| **Abstract by Human**: What sort of battery to use. The industrial designer presented options for materials, components, and batteries and discussed the restrictions involved in using certain materials. |
|---|

| **Longest DA**: well m i'm thinking along the lines of you're you're in the dark watching a dvd and you um you find the thing in the dark and you go like this. |
|---|
| **Centroid DA**: can we uh power a light in this? |
| **Our Method**: |
| - power a light, a strong enough battery, |
| - requires power, a form, |
| - a really good battery, battery power, |
| - watching a dvd, the volume button, lights up or something |

Figure 2: Sample summaries output by different systems for a meeting clip from AMI corpus (less relevant utterances in between are removed). Salient phrases by our system output are displayed for each turn of the clip, with duplicated phrases removed for brevity.

For content features, main speaker tends to utter more salient content. Higher TF-IDF scores also indicate important phrases. For discourse features, structure features matter the most. For instance, jointly modeling the discourse relation of the parent node along with the current node can lead to better inference. An example is that giving more details on the proposal (ELABORATION) tends to lead to POSITIVE feedback. For joint features, features that composite "phrase mentioned in previous turn" and relation POSITIVE feedback or REQUEST yield higher weight, which are indicators for both key phrases and discourse relations. We also find that main speaker information composite with ELABORATION and UNCERTAIN are associated with high weights.

**Error Analysis and Potential Directions.** Taking a closer look at our prediction results, one major source of incorrect prediction for phrase selection is based on the fact that similar concepts might be expressed in different ways, and our model predicts inconsistently for different variations. For example, participants use both "thick" and "two centimeters" to talk about the desired shape of a remote control. However, our model does not group them into the same cluster and later makes different predictions. Furthermore, identifying discourse relations in dialogues is still a challenging task. For instance, "I wouldn't choose a plastic case" should be labeled as OPTION EX-

CLUSION, if the previous turns talk about different options. Otherwise, it can be labeled as NEGA-TIVE. Therefore, models that better handle semantics and context need to be considered.

## 6 Predicting Consistency of Understanding

In this section, we test whether our joint model can be utilized to predict the consistency among team members' understanding of their group decisions, which is defined as consistency of understanding (COU) in Kim and Shah (2016).

Kim and Shah (2016) establish gold-standard COU labels on a portion of AMI discussions, by comparing participant summaries to determine whether participants report the same decisions. If all decision points are consistent, the associated topic discussion is labeled as *consistent*; otherwise, the discussion is identified as *inconsistent*. Their annotation covers the AMI-SUB dataset. Therefore, we run the prediction experiments on AMI-SUB by using the same annotation.[3] Out of total 129 discussions in AMI-SUB, 86 discussions are labeled as consistent and 43 are inconsistent.

We construct three types of features by using our model's predicted labels. Firstly, we learn two versions of our model based on the "consistent" discussions and the "inconsistent" ones in the training set, with learned parameters $\mathbf{w_{con}}$ and $\mathbf{w_{incon}}$. For a discussion in the test set, these two models output two probabilities $p_{con} = \max_{\mathbf{c,d}} P(\mathbf{c}, \mathbf{d}|\mathbf{x}, \mathbf{w_{con}})$ and $p_{incon} = \max_{\mathbf{c,d}} P(\mathbf{c}, \mathbf{d}|\mathbf{x}, \mathbf{w_{incon}})$. We use $p_{con} - p_{incon}$ as a feature.

Furthermore, we consider discourse relations of length one and two from the discourse structure tree. Intuitively, some discourse relations, e.g., ELABORATION followed by several POSITIVE implied consistent understanding.

The third feature is based on word entrainment, which has been shown to correlate with task success for groups (Nenkova et al., 2008). Using the formula in Nenkova et al. (2008), we compute the average word entrainment between the main speaker who utters the most words and all the other participants. The content words in the salient phrases predicted by our model is considered for entrainment computation.

**Results.** Leave-one-out is used for experiments.

---

[3]We thank Joseph Kim for providing the dataset and re-running their system based on our setup.

|  | Acc | F1 |
|---|---|---|
| **Comparisons** | | |
| Baseline (Majority) | 66.7 | 40.0 |
| Baseline (Random) | 50.0 | 32.3 |
| Ngrams (SVM) | 51.2 | 50.6 |
| Kim & Shah | 60.5 | 50.5 |
| **Features from Our Model** | | |
| Consistency Probability (Prob) | 52.7 | 52.1 |
| Discourse Relation (Disc) | 63.6 | 57.1* |
| Word Entrainment (Ent) | 60.5* | 57.1* |
| Prob + Disc+ Ent | **68.2*** | **63.1*** |
| **Oracles** | | |
| Discourse Relation | 69.8 | 62.7 |
| Word Entrainment | 61.2 | 57.8 |

Table 5: Consistency of Understanding (COU) prediction results on AMI-SUB. Results that statistically significantly outperform ngrams-based baseline and Kim and Shah (2016) are highlighted with * ($p < 0.05$, paired $t$-test). For reference, we also show the prediction performance based on gold-standard discourse relations and phrase selection labels.

For training, our features are constructed from gold-standard phrase and discourse labels. Predicted labels by our model is used for constructing features during testing. SVM-based classifier is used. A majority class baseline and a random baseline are constructed as well. We also consider an SVM classifier trained with ngram features (unigrams and bigrams). Finally, we compare with the state-of-the-art method in Kim and Shah (2016), where discourse-relevant features and head gesture features are utilized in Hidden Markov Models to predict the consistency label.

The results are displayed in Table 5. All SVMs trained with our features surpass the ngrams-based baseline. Especially, the discourse features, word entrainment feature, and the combination of the three, all significantly outperform the state-of-the-art system by Kim and Shah (2016).

## 7 Conclusion

We presented a joint model for performing phrase-level content selection and discourse relation prediction in spoken meetings. Experimental results on AMI and ICSI meeting corpora showed that our model can outperform SVM-based classifiers trained with the same feature sets. Further evaluation on the task of predicting consistency-of-understanding in meetings demonstrated that classifiers trained with features constructed from our model's predictions produced superior performance compared to state-of-the-art model.

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
