# Peer review of "Joint Modeling of Content and Discourse Relations in Dialogues"

_ACL 2017 — decision unknown_

[Official Review · Reviewer 1 · rating 3 · confidence 2]
soundness 3 · originality 4 · clarity 4 · impact 4 · substance 3 · appropriateness 5 · meaningful comparison 4 · presentation format Oral Presentation

This paper proposes a joint model of salient phrase selection and discourse
relation prediction in spoken meeting. Experiments using meeting corpora show
that the proposed model has higher performance than the SVM-based classifier.

- Strengths:
The paper is written to be easy to read. Technical details are described fully,
and high performance is also shown in experimental evaluation. It also shows
useful comparisons with related research in the field of discourse structure
analysis and key phrase identification. It is interesting to note that not only
the performance evaluation of phrase selection from discourse, discourse
relation labeling, and summary generation as their applications, but also
application to the prediction of the consistency of  understanding by team
members is also verified .

- Weaknesses:
Jointly Modeling salient phrase extraction and discourse relationship labeling
between speaker turns has been proposed. If intuitive explanation about their
interactivity and the usefulness of considering it is fully presented.

- General Discussion:
SVM-based classifier is set as a comparative method in the experiment. It would
be useful to mention the validity of the setting.